# Novel Diagnostic Options without Contrast Media or Radiation: Triggered Angiography Non-Contrast-Enhanced Sequence Magnetic Resonance Imaging in Treating Different Leg Venous Diseases

**DOI:** 10.3390/diagnostics10060355

**Published:** 2020-05-29

**Authors:** Chien-Wei Chen, Yuan-Hsi Tseng, Chien-Chiao Lin, Chih-Chen Kao, Min Yi Wong, Bor-Shyh Lin, Yao-Kuang Huang

**Affiliations:** 1Institute of Medicine, Chung Shan Medical University, Taichung 402, Taiwan; chienwei33@gmail.com; 2Department of Diagnostic Radiology, Chang Gung Memorial Hospital, Chiayi Branch, Chiayi 61363, Taiwan; 3College of Medicine, Chang Gung University, Taoyuan 333, Taiwan; 4Division of Thoracic and Cardiovascular Surgery, ChiaYi Chang Gung Memorial Hospital, Chiayi and Chang Gung University, College of Medicine, Taoyuan 61363, Taiwan; 8802003@cgmh.org.tw (Y.-H.T.); cvsdoctor@outlook.com (C.-C.L.); cckaomd@gmail.com (C.-C.K.); 5Wound and Vascular Center, Chang Gung Memorial Hospital, Chiayi 61363, Taiwan; 6Institute of Imaging and Biomedical Photonics, National Chiao Tung University, Tainan 71150, Taiwan; mynyy001@gmail.com (M.Y.W.); borshyhlin@gmail.com (B.-S.L.)

**Keywords:** MRI, non-contrast, venography, TRANCE, static ulcer, venous disease

## Abstract

Objectives: Venous diseases in the lower extremities long lacked an objective diagnostic tool prior to the advent of the triggered angiography non-contrast-enhanced (TRANCE) technique. Methods: An observational study with retrospective data analysis. Materials: Between April 2017 and June 2019, 66 patients were evaluated for venous diseases through TRANCE-magnetic resonance imaging (MRI) and were grouped according to whether they had occlusive venous (OV) disease, a static venous ulcer (SU), or symptomatic varicose veins (VV). The clinical appliance of TRANCE-MRI was analysed by groups. Results: In total, 63 patients completed the study. TRANCE-MRI could identify venous thrombosis, including that of the abdominal and pelvic vessels, and it enabled the timely treatment of underlying diseases in patients with OV disease. TRANCE-MRI was statistically compared with the duplex scan, the gold standard to exclude deep vein thrombosis (DVT) in the legs, with regard to their abilities to detect venous thrombosis by using Cohen’s kappa coefficient at a compatible value of 0.711. It could provide the occlusion degree of the peripheral artery for treating an SU. Finally, TRANCE-MRI can be used to outline all collateral veins and occult thrombi before treating symptomatic or recurrent VV to ensure a perfect surgical plan and to avoid complications. Conclusions: TRANCE-MRI is an innovative tool in the treatment of versatile venous pathology in the lower extremities and is widely used for vascular diseases in our institution.

## 1. Introduction

Venous diseases in the lower extremities include minor varicose veins and static ulcers (SUs), ranging from ambulatory venous hypertension to potentially deadly occlusive diseases, such as deep vein thrombosis (DVT) [1,2,3]. Currently, only a few options are available for the objective venous evaluation of the lower limbs [4,5,6,7]. The venous system is not exactly enhanced by the computed tomography (CT) venogram, and a high-quality enhancement requires specific access (from a morbid limb). Compared with conventional angiography, most of the magnetic resonance venography (MRV) techniques involving contrast media have been proven to be highly sensitive for detecting lesions in vessels [8,9]. Although magnetic resonance imaging (MRI) involves the use of contrast agents instead of radiation exposure, it still has undesirable effects. The occurrence of nephrogenic systemic fibrosis (NSF) is a rare but a dangerous complication resulting from the use of gadolinium-based agents in patients with poor kidney function [10]. The triggered angiography non-contrast-enhanced (TRANCE) technique records differences in vascular signal intensity during the cardiac cycle for subsequent image subtraction and provides vascular images without using contrast agents. However, few clinical applications of this technique are known to exist for imaging the vascular pathology of the lower extremities [9,11,12]. In this study, we present our experience of using TRANCE-MRI and indicate its strengths and weaknesses in different cases of venous disease in the legs.

## 2. Methods

### 2.1. Patients

This study was approved by the Institutional Review Board (IRB) of the Chang Gung Memorial Hospital (IRB numbers: 201802137B0 and IRB 201901058B0, approved on 8 June 2019), and informed consent was obtained from all subjects. The study was performed in accordance with the relevant guidelines and regulations.

The study included patients who had been evaluated through TRANCE-MRI for venous pathology in their lower extremities at a tertiary hospital between April 2017 and June 2019. We prospectively collected and retrospectively analysed the clinical values of their data. Furthermore, venous pathology in the lower extremities was suspected in all patients. Exclusion criteria were pregnancy and having non-MRI-compatible ferromagnetic devices. In addition, patients were excluded if they exhibited poor compliance or had multiple comorbidities that prevented them from lying down for the 1 h protocol of TRANCE-MRI. Initially, 66 patients were evaluated. One lady was excluded due to possible pregnancy, and another three patients were either morbidly obese or restless, rendering them unable to continue with the MRI study. All the patients underwent a non-invasive colour Doppler examination in the supine position before scheduled TRANCE-MRI. The femoral veins, great saphenous vein (GSV), popliteal veins (PV), and perforating vein in the calves were examined. Pelvic veins were not evaluated in Doppler examinations. Other imaging modalities, such as the computed tomographic venogram (CTV) and lymphoscintigraphy by Tc-99m, were performed according to specific indications.

### 2.2. MRI Acquisition

MRI was performed using a 1.5 T MR scanner (Philips Ingenia, Philips Healthcare, Best, Netherlands). Patients underwent imaging in the supine position and with a peripheral pulse unit trigger. All the images of arterial systems were evaluated by a three-dimensional (3D) turbo spin-echo (TSE) during systole and diastole periods. TSE TRANCE imaging was performed using the following parameters: repetition time (TR), 1 beat; echo time (TE), shortest; flip angle, 90°; voxel size, 1.7 × 1.7 × 3 mm^3^; and field of view (FOV), 350 × 420. During systole, the arterial blood flow is relatively fast, causing the dephasing of the signal and leading to flow voids; hence, arteries appeared black with systolic triggering. During diastole, the blood flow in arteries is slow; hence, signal dephasing did not occur and the arteries appeared bright in diastolic scans. The subtraction of two phased scans comprised a 3D data set with only arteries. Other images of venous systems were evaluated through 3D TSE Short tau inversion recovery (STIR) during the systole period. TSE STIR TRANCE imaging was performed using the following parameters: TR, 1 beat; TE, 85; inversion recovery delay time, 160; voxel size, 1.7 × 1.7 × 4 mm^3^; and FOV, 360 × 320. STIR provides extra background suppression because the signals of fat and bones are also suppressed. The arteries appeared black with systolic triggering. The TRANCE-MRI result of the venous system was a data set, in which no subtraction was required [13]. A quantitative flow scan was routinely performed to determine the appropriate trigger delay times for systolic and diastolic triggering [14]. All the images were acquired without using the gadolinium contrast medium. The TRANCE-MRI protocol required 40 min for image acquisition, 25 min for MRV, and 35 min for magnetic resonance arteriography. In addition to the peripheral pulse unit or electrocardiogram (ECG) triggering, the TRANCE-MRI technique was routinely used to perform a quantitative flow (Q-Flow) scan of the abdominal aorta above the aortic bifurcation to determine the appropriate trigger delay times for systolic and diastolic triggering. Therefore, the pulse wave damping in the peripheral extremity had little effect on inappropriate triggering times. After the TRANCE MRI was completed, an experienced radiologist and cardiovascular surgeon annotated each image using a 4-point grading scale. The image quality of the TRANCE-MRI was analysed based on noise, resolution, and homogeneity, grading it as perfect, good, acceptable, or bad, in order. Image artefacts were also analysed and scored as not present, not disturbing, disturbing, and not applicable. The image quality was stable between observers (Figure 1).

### 2.3. Statistical Analysis

The continuous variable (age) was analysed using an unpaired two-tailed Student’s *t* test or one-way analysis of variance test, and the discrete variables (sex, substance usage, comorbidities, interventions, and TRANCE MR findings) were compared using a two-tailed Fisher’s exact test. All the statistical analyses were conducted using the STATA statistics/Data Analysis 8.0 software (Stata Corporation, College Station, TX, USA). The results are presented as means and standard deviations. Statistical significance was defined as *p* < 0.05.

## 3. Results

The demographic characteristics (gender, age, substance use, comorbidities, and history of venous surgery) of 63 patients are summarised in Table 1. The patients were classified into three groups—occlusive venous (OV) disease, SU, and varicose veins (VV)—based on the status of their veins at presentation. The OV group comprised 35 patients who presented with typical clinical features (sudden leg swelling, recent history of trauma, and surgery) and symptoms during physical examination (dark skin colour, pitting oedema, and no cracking/flaking of the skin). The SU group comprised 12 patients who visited our clinics for chronic unhealed leg wounds (over 3 months), especially in the gaiter area. The VV group comprised 16 patients who had painful or recurrent VV after venous intervention. The comparison between the descriptive statistics of these groups is also presented in Table 1. Patients in the VV group were younger, primarily women, and had fewer comorbidities. Patients in the OV group were older and had more comorbidities, especially with regard to malignant disease and renal function impairment.

In the OV group, the symptoms were caused by DVT in 20 patients (20/35, 57%; Table 2). Most of the patients received an oral anticoagulant, and 11 of them required admission for heparinisation. An inferior vena cava (IVC) filter was placed in two patients for preventing pulmonary embolism, and one patient received a catheter-based thrombolytic therapy by using either urokinase or an EkoSonic endovascular system (EKOS; BTG Interventional Medicine, Bothell, WA) to salvage the threatened limb (Figure 2). Notably, the OV diseases caused by DVT were associated with malignant diseases (hepatic cellular carcinoma, lung cancer, prostate cancer, and laryngeal cancer) in four patients (4/20, 20%). The patient with a final diagnosis of advanced prostate cancer was initially detected with diffused adenopathy and pelvic venous occlusion by TRANCE-MRI. By contrast, among 35 patients who presented with OV disease, 15 were free from DVT, as indicated by TRANCE-MRI. The OV diseases were caused by either external compression of benign aetiology (myositis, iliac artery aneurysm, hip osteomyelitis, and Baker’s cyst) or malignant diseases, such as rhabdomyosarcoma. Furthermore, TRANCE-MRI clarified the lower leg arterial system and its relationship with pelvic veins. The TRANCE-MRI of the arterial system revealed peripheral arterial diseases in three patients and an iliac arterial aneurysm in one patient. Interestingly, the TRANCE-MRI identified 13 patients (13/35, 37.1%) with features typical of May–Thurner syndrome (venous stricture by arterial compression). For two patients, the therapeutic strategy was changed to address the malignancy after the TRANCE MR evaluation. Meanwhile, we discontinued the oral anticoagulation in two patients of the OV group without DVT, after it was proven there were no thrombi in their venous systems. In summary, OV disease caused by DVT has a quicker onset and greater requirements for anticoagulation therapy, and duplex/TRANCE-MRI facilitates in detecting more thrombi. TRANCE-MRI was statistically compared with the duplex scan (the gold standard to exclude DVT in the legs) with regard to their abilities to detect venous thrombosis using Cohen’s kappa coefficient at a value of 0.711.

Table 3 summarises the data of 12 patients who presented with SUs for more than 3 months. Of these 12 patients, four patients received either truncal vein ablation or stripping and two patients received soft tissue reconstruction by either skin grafting or free flap transfer. Their wounds were shallow with no bone exposure. TRANCE-MRI evaluations revealed that the SUs of four patients may have been attributable to post-thrombotic sequelae with residual thrombi, those of one patient were caused by external compression by joint fluids, and those of three patients were attributable to profound VV from GSV and perforating branches. Five patients presented with normal venous anatomy with diffuse soft tissue swelling, which favours lymphoedema. Peripheral arterial occlusions were revealed by TRANCE-MRI in two patients, which were not related to static foot ulcers. Five patients with venous thrombosis received oral anticoagulants for SUs with great responses.

TRANCE-MRI was scheduled for 16 patients in the VV group for their recurrent or symptomatic VV. The indicators for advanced venous images are listed in Table 4. The major reasons were found to be atypical clinical images or recurrent VV with previous venous surgery (six patients). Two patients underwent truncal ablation with phlebectomy guided by TRANCE-MRI. Notably, TRANCE-MRI indicated that one patient had occult DVT in the left popliteal vein with a compressed left common iliac vein (Figure 3), and another patient had Klippel–Trenaunay syndrome. No surgical interventions were performed in these two patients.

## 4. Discussion

Venous pathology in the lower extremities may be considered in patients with tortuous calf veins, “gaiter” wounds, and asymmetrical legs. Ultrasonography (US), as a screening tool for venous diseases, is operator-dependent and inadequate for providing information about pelvic vessels [4,15,16]. Conventional venography has been considered the gold standard for detecting DVT in legs. However, it is invasive and uses radiation and contrast media. Although CT venography is useful for excluding pulmonary embolism in patients with signs of thrombosis in the legs, it cannot replace US as a means of first-line imaging for detecting DVT in the legs [17,18]. Time-of-flight (TOF)-MRV is less invasive than conventional and CT venographies and prevents the side effects of iodinated contrast material. The disadvantage of TOF-MRV is that the FOV is small for each obtained image and that it is time consuming for obtaining a full image of the lower extremity [8]. MRI with gadolinium-based contrast media is a quicker method for imaging the lower extremities [19,20,21,22]. Although MRI does not involve radiation exposure, the noniodinated contrast agents that are used have harmful effects. NSF is a dangerous complication of gadolinium-based contrast agents in patients with pre-existing impairments of kidney function. However, it may even occur in patients with normal renal function [23,24]. The TRANCE technique in MRI was first publicised in 1985 [25]. Although TRANCE-MRI has been used for diagnosing cranial diseases and mostly arterial diseases, there have been only a few applications of this technique in assessing venous pathology for the lower extremities. [11,12,14,26].

The principle of TRANCE-MRI is that different velocities of blood flow have different signal intensities on the TSE sequence. A high signal intensity and bright colour reflect slow velocity such as that in venous blood flow and diastolic arterial blood flow. The high velocity of systolic arterial blood flow results in a flow void effect, with dark colour and low signal intensity. High-resolution images and image-isolated vascular structures, such as arteries and veins, can be obtained through TRANCE-MRI [14]. An image presenting only the venous structure without the accompanying arterial structure is difficult to achieve through MRI or CT using contrast media because the proper acquisition time is short and variable. Therefore, TRANCE-MRI is a useful tool for examining the venous pathology of the lower extremities because it provides additional information regarding pelvic vessels, requires no contrast agents, and involves no radiation toxicity. Our preliminary experience highlights that TRANCE-MRI may be safe and useful for imaging the lower extremities, especially in venous pathology [14]. TRANCE-MRI would be particularly suitable for patients with chronic renal insufficiency, a history of abdominal/pelvic/orthopaedic surgery, or allergies to contrast media [13].

Several advantages of TRANCE-MRI application in venous pathology in the lower extremities have been demonstrated. First, TRANCE-MRI provides images of not only the arteries and veins in the lower extremities but also information concerning the pelvis and abdomen for patients with DVT. DVT may be confused as the external compression of the pelvic vessels, and it is also a sign of occult malignancies. Second, thrombi and collateral veins can be clearly outlined, including middle femoral veins that might be difficult to detect through US. This can aid in catheter-based thrombolytic therapy/EKOS and rescue therapy in recurrent VV after truncal ablations of the GSV. Finally, because TRANCE-MRI does not emit radiation or involve the use of contrast media, it is safe for patients with impaired renal function.

In this study, we summarised the data of our 63 patients who received TRANCE-MRI and attempted to determine their unique value for different venous pathologies (OV, SU, and VV). The appliance of TRANCE-MRI plays a crucial role in the therapeutic algorithm in our vascular wound centre now (Figure 4). TRANCE-MRI provided accurate information on DVT as well as the anatomic structure for patients who presented with OV. In addition, it objectively guided the catheter-based thrombolysis/venous angioplasty/venous stenting. TRANCE-MRI can distinguish uncomplicated congenital venous variations, benign external compression (joint fluid and aortic aneurysms), and adjacent malignant compression (metastatic lymph nodes and retroperitoneal cancers) from DVT. It enables the timely treatment of underlying diseases without the unnecessary use of anticoagulants.

TRANCE-MRI can provide not only the results of prior venous surgery but also the details of the occlusion degree of foot arteries for patients with SUs. This information can enable physicians to decide on reinterventions for venous problems and appropriately administer anticoagulation therapy to the correct patients [27,28]. On the other hand, compressive therapy with full arterial mapping may be safe for SUs.

A duplex scan may be insufficient for treating complicated VV (cellulitis and atypical presentation) and recurrent VV (the VV group) after previous GSV ablation. As detailed in Table 4, six patients (6/16, 37.5%) received greater saphenous vein stripping or ablation. TRANCE-MRI provided objective mapping for evaluating prior surgical results and the potential causes of symptoms. Notably, one patient was proven to have hidden right common iliac venous thrombosis; hence, their GSV laser ablation was cancelled. TRANCE-MRI could clearly demonstrate the major collaterals in addition to saphenous veins and aid in the formulation of an exact surgical plan for less invasive venous ablation.

Compared with TRANCE-MRI, US has a smaller role in assessing VV of the lower extremities and deep veins of the pelvis and abdomen. TRANCE-MRI and the duplex scan were found to have comparable detection abilities for DVT based on the shared kappa coefficient value of 0.7. However, we still consider that US should be used preferentially for assessing venous lesions in the lower extremities because it is non-invasive and cost effective. TRANCE-MRI is a non-invasive examination that does not use a contrast medium and provides images of arteries and veins in the lower extremities as well as information regarding the pelvis and abdomen. If a patient experiences pelvic vein problems or complicated VV before surgery, we recommend using TRANCE-MRI.

This study identified some drawbacks of TRANCE-MRI. First, TRANCE-MRI of the venous system is less clear in the pelvic vessels, which could be attributed to the complex anatomy and overlapping of vessels with different blood flow directions. Other observations such as constant filling defects, the increased diameter and number of collateral veins, and the application of intravascular ultrasound may reduce the incorrect diagnosis risk. Second, structures and lesions (e.g., intramural hematomas) presenting no differences in signal intensity during the cardiac cycle will be obscured during the subtraction process to generate the 3D MRA data set. Third, vascular pulsation has little effect on motion artefacts in peripheral MRA. However, gross motion exerts an insurmountable effect on motion artefacts during imaging acquisition. Thus, this protocol is not suitable for critical and irritable patients. Fourth, different severities of arrhythmia have variable effects on the correct acquisition time. Fifth, partial blurring due to T2-decay may occur during the rapid FSE echo train [29]. The most spatial blurring occurs when TE is short because the higher order phase encoding steps that provide edge detail are being filled with late echoes in the train. Thus T1- and proton-density-weighted images present the highest degree of blurring. Sixth, the TRANCE-MRI protocol requires 60 min for imaging (25 min for MRV and 35 min for MRA). Finally, TRANCE-MRI is expensive and not yet readily available at some institutions.

### Study Limitations

The major limitations of this study were as follows: the non-randomised design, the small sample size, and a lack of the comparison of interobserver variability and adequate validation with other imaging studies. The other weakness is a lack of the solid comparison of the quantitative and qualitative analyses of image quality. It is difficult to perform image quality assessment between ultrasonography and TRANCE-MRI. Instead of image quality assessment, we applied Cohen’s kappa test to measure inter-rater reliability between ultrasonography and TRANCE-MRI. However, we provide our therapeutic algorithm of the vascular centre, for the other institutions whilst starting their TRANCE-MRI service. This study used TRANCE-MRI for clinically assessing different venous pathologies. TRANCE-MRI may provide more useful information for treating complicated lower venous diseases.

## 5. Conclusions

TRANCE-MRI provided images of arteries and veins in the lower extremities and information about the pelvis and abdomen. TRANCE-MRI could identify venous thrombosis, including that of the abdominal and pelvic vessels, and enabled the timely treatment of underlying diseases without the requirement for anticoagulants. Furthermore, it could reveal the occlusion degree of the peripheral artery to treat SUs with a safe compression therapy and outlined all collateral veins and occult thrombi before the treatment of symptomatic VV to ensure the selection of the optimal surgical plan. TRANCE-MRI is a powerful tool in the treatment of venous pathology in the lower extremities and will be widely used in the near future.

## Figures and Tables

**Figure 1 diagnostics-10-00355-f001:**
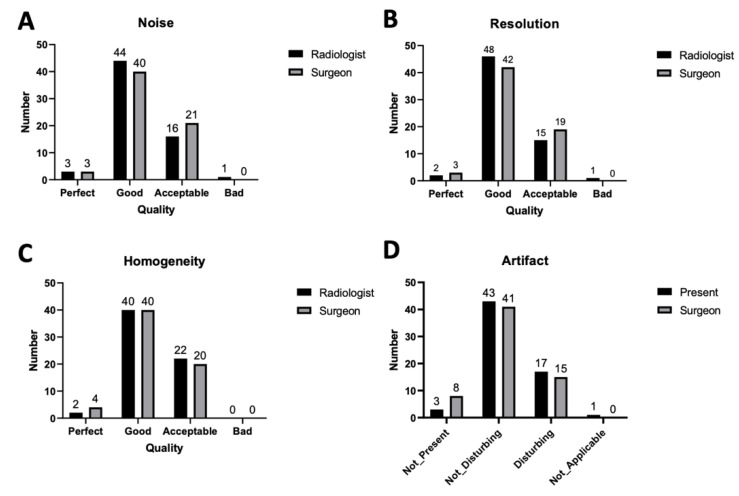
Image quality comparison, including (**A**) noise, (**B**) resolution, (**C**) homogeneity, and (**D**) artefacts.

**Figure 2 diagnostics-10-00355-f002:**
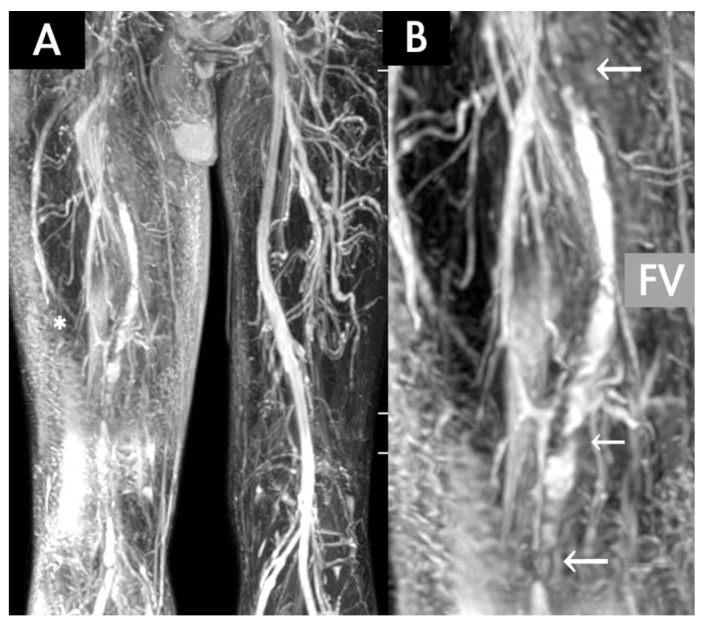
A 51-year-old man with acute deep vein thrombosis of the right lower limb who received an EkoSonic endovascular system (EKOS) thrombolytic therapy to the counter-lateral limb 3 days ago. (**A**) TRANCE-MRI revealing diseased venous systems and extensive soft tissue (white asteroid) enhancement in the morbid right thigh and normal veins in the left thigh. (**B**) Multiple residual venous thrombi (white arrows) in the right femoral vein (FV).

**Figure 3 diagnostics-10-00355-f003:**
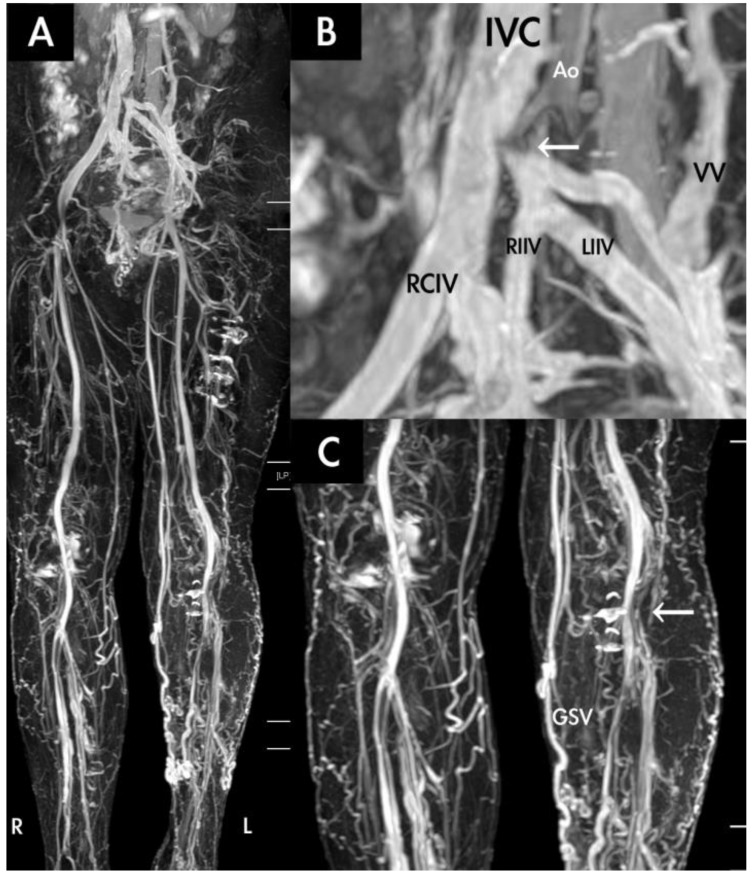
A 60-year-old woman who received TRANCE-MRI for her painful left varicose vein with 50 m claudication. (**A**) Her greater saphenous vein in left leg was obviously more engorged than that in the right leg. (**B**) Normal inferior vena cava (IVC) and right common iliac vein. Her right internal iliac vein and left internal iliac vein drain into the left common iliac vein (white arrow), which is compressed by the aorta (Ao). Prominent collateral veins (VV) draining into the renal vein can be clearly observed. RCIV: right common iliac vein; RIIV: right internal iliac vein; LIIV: left internal iliac vein (**C**) A diseased left popliteal vein (white arrow) and varicose left greater saphenous vein (GSV) can be observed.

**Figure 4 diagnostics-10-00355-f004:**
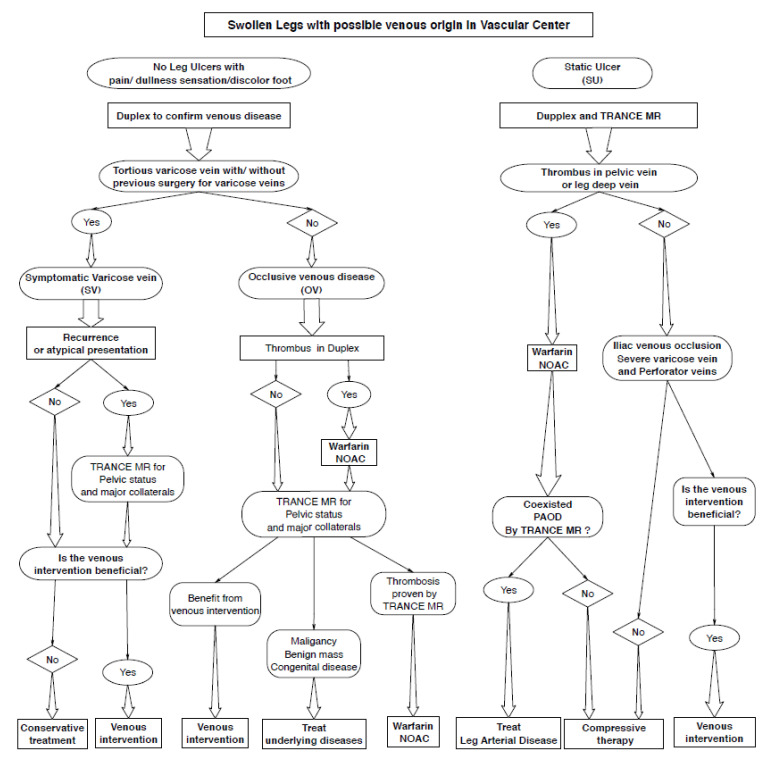
Therapeutic algorithm in the vascular centre—application of TRANCE-MRI for the lower extremity with venous origin. NOAC: non-warfarin oral anticoagulation.

**Table 1 diagnostics-10-00355-t001:** Demographics characteristics of patients with three different presentations for lower leg venous disease.

	Occlusive Venous Symptoms (OV)	Venous Static Ulcer (SU)	Symptomatic Varicose Vein (VV)	Sum (%)
Total	35	12	16	63
Male gender (%)	19 (54%)	11 (92%)	3 (18%)	33 (52%)
Age (year)	65.5 ± 12.9	59.2 ± 11.6	52.3 ± 15.9	61 ± 14.6
Substance use				
Smoking	6	3	1	10 (15.9%)
Alcohol	7	2	2	11 (17.5%)
Betel nuts	3	3	1	7 (11.1%)
Comorbidities				
Hypertension	18	3	3	24 (38.1%)
Diabetes mellitus	11	4	2	17 (27%)
CAD	2	0	0	2 (3.2%)
Stroke	1	0	0	1 (1.6%)
Cancer	9	1	0	10 (15.9%)
Chronic renal failure	2	0	1	3 (4.8%)
Hemodialysis	2	0	0	2 (3.2%)
Previous venous surgery				
Stripping	0	0	2	2 (3.2%)
GVS ablation	0	3	4	7 (11.1%)
Sclerotherapy	0	1	0	1 (1.6%)
IVC filter	2	0	0	2 (3.2%)
Pelvic/orthopaedic	6	0	0	6 (9.5%)

CAD: coronary arterial disease; GSV: greater saphenous vein; IVC: inferior vena cava.

**Table 2 diagnostics-10-00355-t002:** Details of venous obstructive syndrome (*n* = 35 patients).

	DVT	Non DVT	*p* Value
Subgroup patient numbers	20	15	
Age (years old)	64.8 ± 10.5	66.53 ± 16.2	
Male gender	12	7	0.506
Onset less than 8 weeks	10	2	0.034 *
Duplex in leg suspected for thrombi	17	2	<0.001 *
TRANCE MR in vein			
**Deep vein** thrombus	20	0	<0.001 *
Congenital anomaly	2	1	1
May–Thurner-like (arterial compression)	10	3	0.89
Malignant disease in MRI	4	3	1
External compression, malignant	1	3	0.292
External compression, benign	2	4	0.367
Pelvic congestion	1	2	0.565
TRANCE MR in artery			
PAOD	2	0	0.496
AAA and IAAA	0	1	1
CTA and CTV	3	1	0.619
Lymphoscintigraphy	1	0	1
Intervention			
CDT and EKOS	2	0	0.496
Heparinisation in hospital	11	1	0.004 *
NOAC or warfarin	19	2	<0.001 *
Venous angioplasty	4	1	0.365
Venous stenting	1	0	1
IVC filter	2	0	0.496
Clinical			
New neoplasm diagnosis <1 year	3	3	1
Pulmonary emboli	2	0	0.496

AAA: abdominal aortic aneurysm; CDT: catheter-directed thrombolysis; CTA: computed tomography angiography; CTV: computed tomographic venogram; EKOS: EkoSonic endovascular system; IAAA: iliac arterial aneurysm; IVC: inferior vena cava; NOAC: non-warfarin oral anticoagulation; PAOD: peripheral arterial occlusive disease; TRANCE-MRI: triggered angiography non-contrast-enhanced magnetic resonance imaging. * asteroid indicates significance.

**Table 3 diagnostics-10-00355-t003:** Demographic characteristics of 12 patients with leg static ulcers.

Age (Years)	59.3 ± 12.2
Male gender	11 (92%)
Doppler screening	
Valvular insufficiency	2
Venous thrombi	5
TRANCE MR in vein	
Deep vein thrombus	4
Congenital anomaly	0
May–Thurner-like picture	1
Malignant disease	1
Profound varicose vein	10
Pelvic congestion	2
Subcutaneous tissue enhancement without venous pathology	5
External compression (joint fluid)	1
TRANCE MR in artery	
PAOD	2
Intervention history	
NOAC or warfarin	5
Skin graft/free flap	2
Venous ablation/stripping	4

NOAC: non-warfarin oral anticoagulation; PAOD: peripheral arterial occlusive disease; TRANCE-MRI: triggered angiography non-contrast-enhanced magnetic resonance imaging.

**Table 4 diagnostics-10-00355-t004:** Demographic characteristics of 16 patients with symptomatic varicose veins.

Age (Years)	52.3 ± 15.9
Male gender	3 (18%)
Why do the TRANCE MR in the vein?	
Rapid progression of the varicose vein	3
Claudication	5
Phlebitis, cellulitis, and bleeding episodes	4
Recurrence of the varicose vein after interventions	4
Doppler study in leg veins	
Superficial venous thrombosis	3
Deep venous thrombosis	0
Valve incompetence	11
TRANCE MR in vein	
Thrombus in deep venous system	1
Congenital anomaly	1
May–Thurner-like picture	1
Malignant disease	1
TRANCE MR in artery	
PAOD	0
Intervention history	
NOAC or warfarin	3
Venous ablation/stripping	6
Venous operation after TRANCE MR	
Truncal ablation with phlebectomy	2

NOAC: non-warfarin oral anticoagulation; PAOD: peripheral arterial occlusive disease; TRANCE-MRI: triggered angiography non-contrast-enhanced magnetic resonance imaging.

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
