# Peer review of "Novel Diagnostic Options without Contrast Media or Radiation: Triggered Angiography Non-Contrast-Enhanced Sequence Magnetic Resonance Imaging in Treating Different Leg Venous Diseases"

_diagnostics, 2020, doi:10.3390/diagnostics10060355_

Round 1

Reviewer 1 Report

Abstract should be rearranged for better understanding on the exact goal of the study.

Table 1 and 2 should be combined.

Some minor spelling errors.

Please provide more detailed information about results with one table summarizing quantitative and qualitative analysis of image quality (this is missing).

The absence of comparison with other imaging modality is a weakness.

Author Response

Reply to Reviewer 1

Comment 1

Abstract should be rearranged for better understanding on the exact goal of the study.

Reply Comment 1

We removed redundant description in the abstract in this version.

Comment 2

Table 1 and 2 should be combined.

Reply comment 2

We had combined the table 1 and 2 in this revise version.

Comment 3:

Some minor spelling errors.

Reply comment 3

We corrected some minor spelling errors. (“saphenous” in line 277, noninvasive non-invasive in line 287)

Comment 4

Please provide more detailed information about results with one table summarizing quantitative and qualitative analysis of image quality.

The absence of comparison with other imaging modality is a weakness.

Reply to Comment 3

Indeed, it is the major weakness of this study. Ideally, this article may provide more evidence with the quantitative and qualitative analysis of image quality including CT venogram and TRANCE MR.

However, CT venograms are not available in all attendants in this study. CT venogram is potentially harmful to those patients with impaired renal function. Moreover, the CT venogram may require delivering the contrast media from the distal part of the morbid limb for the best venous imaging. That is difficult in the patients with static venous ulcer. Thus, we could not perform quantitative and qualitative analysis of image quality between TRANCE MRI and CT venogram.

Instead of image quality assessment, we apply cohen's kappa test to measure inter-rater reliability between ultrasonography and TRANCE-MRI. Meanwhile, we provide an imaging quality analysis in “method” section (line 99-103) with a new figure.

We response this comment in the “study limitation”. line 301 to line 304, study limitation as follow.

“The other weakness is lacking solid comparison quantitative and qualitative analysis of image quality. It is difficult to perform image quality assessment between ultrasonography and TRANCE-MRI. Instead of image quality assessment, we apply cohen's kappa test to measure inter-rater reliability between ultrasonography and TRANCE-MRI.”

Sincerely,

Yao-Kuang Huang, MD, PhD

Division of Thoracic and Cardiovascular Surgery

Chia-Yi Chang Gung Memorial Hospital

Fax: 886-975368209

E-mail: huang137@icloud.com

Reviewer 2 Report

Although small, non-randomized and without controls the study has been well performed, nicely presented with excellent images and tables in a well prepared manuscript.

Author Response

We deeply appreciate your kindness. 

Sincerely,

Yao-Kuang Huang and Chieh-Wei Chen

Division of Thoracic and Cardiovascular Surgery

Chia-Yi Chang Gung Memorial Hospital

Fax: 886-975368209

E-mail: huang137@icloud.com